# Mineralogy and Geochemistry of Sr-Bearing Phosphates from the Nanping No. 31 Pegmatite (SE China): Implications for Sr Circulation and Post-Magmatic Processes in Granitic Systems

**Can Rao [1],\*, Ru-Cheng Wang [2], Frédéric Hatert [3], Run-Qiu Wu [1] and Qi Wang [1]**

[1]  Key Laboratory of Geoscience Big Data and Deep Resource of Zhejiang Province, School of Earth Sciences, Zhejiang University, Hangzhou 310027, China; 21838006@zju.edu.cn (R.-Q.W.); 21938007@zju.edu.cn (Q.W.)

[2]  State Key Laboratory for Mineral Deposits Research, School of Earth Sciences and Engineering, Nanjing University, Nanjing 210046, China; rcwang@nju.edu.cn

[3]  Laboratoire de Minéralogie, B18, Université de Liège, B-4000 Liège, Belgium; FHatert@ulg.ac.be

\*  Correspondence: canrao@zju.edu.cn

**Abstract:** Although Rb/Sr ratio and Sr isotopes are routinely modeled to determine petrogenetic processes and sources for granitic systems, the post-magmatic path and crystallization of Sr in granitic systems have not been thoroughly elucidated to date. In this study, we present the petrography, chemical composition and isotopic $^{87}Sr/^{86}$ of Sr-bearing phosphates from the Nanping No. 31 pegmatite in Southeastern China, helping to characterize post-magmatic stages and geochemical recirculation of Sr in granitic systems. K-feldspar and primary apatites occur as major "primary Sr minerals", the occurrences of secondary Sr phosphates (strontiohurlbutite, palermoite and goyazite) and Sr-rich phosphates (apatites, hurlbutite, bertossaite and fluorarrojadite-(BaNa)) reflect the transport, concentration and recrystallization of Sr in granitic systems. The mobilization and recrystallization of Sr in granitic systems are mainly controlled by the variation in alkalinity of hydrothermal fluids. Two post-magmatic recirculations of Sr are proposed in the Nanping No. 31 pegmatite: (1) breakdown of the "primary Sr mineral" (K-feldspar and primary apatites) and crystallization of secondary Sr-bearing phases; and (2) replacement of secondary Sr-bearing phosphates and direct precipitation of later palermoite and goyazite from later Sr-rich fluids at low temperatures. The Sr isotope features of Sr-bearing phosphates suggest that the emplacement and consolidation processes of the Nanping pegmatite involved the participation of externally derived fluids.

**Keywords:** phosphate; Sr-bearing minerals; Sr isotope; granitic systems; Nanping pegmatite

## 1. Introduction

Strontium is one of lithophile metallic elements, mainly dispersed as trace element in rock-forming minerals, which resulted in the rare crystallization of Sr minerals [1]. Its isotopic characteristic was widely utilized as a good tracer of element sources in melts/fluids [2–4], as a powerful record of fluid–rock interactions [5–8], and as an important long-term indicator of geological processes [1,9]. The geochemical behavior of Sr in granitic systems shows a coherence with Ca, and Sr prefers to concentrate in late crystallization phases [10,11]. Sr can be redistributed by hydrothermal fluids during the hydrothermal stages of granitic systems, resulting in the crystallization of secondary Sr phosphates such as palermoite and goyazite, demonstrating the hydrothermal mobility of Sr in granitic systems [3,12–15]. Hydrothermal fluids from granitic systems may be contaminated by external fluids, and secondary Sr-bearing minerals can record "initial" Sr isotopic composition of original rocks during

their crystallization [16]. However, the post-magmatic Sr history is still expected to be reconstructed in highly evolved granitic systems [3,17], and the crystallization and recirculating processes of Sr in granitic systems are poorly understood.

The Nanping No. 31 pegmatite (Southeastern China) is not only highly enriched in Nb, Ta, Sn, Be and Li, but hosts abundant Sr phosphates (e.g., strontiohurlbutite, palermoite and goyazite) and Sr-rich phosphates (e.g., fluorapatite, hydroxylapatite, hurlbutite, bertossaite and fluorarrojadite-(BaNa)). In this study, we investigated the petrography, chemical composition and Sr isotopes of Sr-bearing phosphates from the Nanping No. 31 pegmatite, with the aims to constrain the whole Sr history of granitic systems, on the one hand, and to reveal the post-magmatic alteration of granitic pegmatite, on the other hand. These data will also provide new insight into the mineralogy and geochemistry of Sr, as well as into post-magmatic processes of granitic pegmatite.

## 2. Geological Background and Sampling

The Nanping pegmatite field is located at the southeastern part of the Cathaysian block in Fujian province, Southeastern China (Figure 1a). The Cathaysian block consists of the Proterozoic basement with a cover sequence of Sinian to Triassic sedimentary strata [18]. The Early Paleozoic and Early Mesozoic tectono-thermal events resulted in widespread partial melting of continental crust, and generated large-scale S-type granites with peak ages of 430–400 Ma and 240–200 Ma in the Cathaysian continental blocks [19]. The Northwestern Fujian Province underwent Hercynian–Indosinian movement; only Yanshan movement generated numerous Caledonian and Hercynian pegmatites. Caledonian pegmatite is mostly concentrated in the north–central part of the Caledonian fold belt, while Hercynian pegmatites are limited in the south side of the belt [20]. The Nanping pegmatite is distributed along the southeast margin of Caledonian folded belt in the Northwestern Fujian Province, Southeastern China. In this region, the metamorphic rocks are composed of fine-grained amphibolite facies, garnet- and sillimanite-bearing biotite gneiss, schist and quartzite, which mainly occur in the Dikou Formation. The Dikou Formation formed later than 0.8 Ga, and its age of orogenic metamorphism is younger than 604 Ma [21].

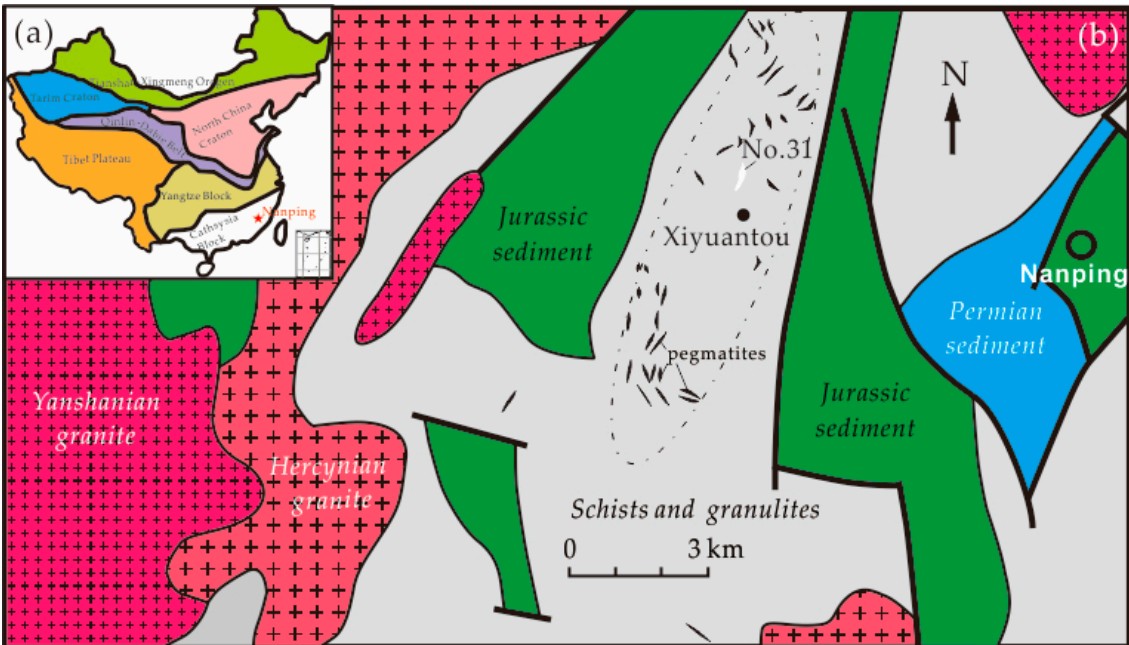

**Figure 1.** Location and simplified geological map of the Nanping pegmatite district. (**a**) Location of the studied area; (**b**) Simplified geological map of the Nanping pegmatite (modified from Reference [20]).

In the Nanping pegmatite field, the exposed strata are divided into four litho-tectonic units: Proterozoic basement unit, Upper Neoproterozoic to Ordovician weakly metamorphosed sediments, a sequence of Upper Paleozoic to Lower–Middle Triassic rocks and Mesozoic–Cenozoic volcanic and sedimentary rocks. The Proterozoic basement unit is composed of Meso- and Neo-Proterozoic schists and granulites of Xiafeng fabric in the Wanqun rock suite, which was intruded by the majority of Nanping pegmatites. Folds and faults are well-developed in the basement rocks, and most of sedimentary rocks were destroyed to discrete fragments by late faults. In this field, the Nanping synclinorium is a first-class tectonic unit, which can be further divided into NE and SW directions by sub-folds, and the later overlapped with the former. Hercynian and Yanshanian (Jurassic) granites (Figure 1b) are the main magmatic events in this pegmatite field, which is characterized by high Si, alkalinity and peraluminosity, but is poor in Ti, Fe, Mg and Ca, and their initial $^{87}Sr/^{86}$ ratios are more than 0.71 [20].

The Nanping No. 31 pegmatite (Figure 1b) is one of the highly enriched rare-element (Nb, Ta, Sn, Be and Li) pegmatites in the Nanping pegmatite field with well-developed mineral-textural zonation and belongs to an Li-Cs-Ta (LCT) type pegmatite [22]. It is 5 to 6 m in width, 90 m in depth and 300 to 600 m in length. From the outermost zone inward, five discontinuous mineral-textural zones were distinguished [20]: quartz-albite-muscovite zone (zone I), saccharoidal albite ± muscovite zone (zone II), albite-coarse quartz-spodumene zone (zone III), quartz-spodumene-montebrasite zone (zone IV) and blocky quartz-K-feldspar zone (zone V). The detailed petrographic and mineralogical features of five zones have been described in the literature [22–24]. Zone I primarily consists of medium-grained quartz and muscovite and fine-grained albite with small amounts of cassiterite, columbite-tantalite, zircon, beryl, hurlbutite, phenakite, apatite and strontiohurlbutite. Zone II is characterized by saccharoidal albite (> 90 vol%), with small amounts of greenish muscovite and quartz; cassiterite, columbite-tantalite, wodginite and tapiolite-(Fe) mainly occur at the boundary between the saccharoidal albite and greenish muscovite. Beryl, phenakite, hydroxylherderite, hurlbutite, euclase and strontiohurlbutite are also distributed in this zone. Zone III mainly contains platy crystals of albite, coarse-grained quartz and spodumene, with lesser amounts of columbite-tantalite, wodginite-group minerals, tapiolite, microlite, zircon, cassiterite, apatite, montebrasite and beryl. Zone IV is mainly composed of coarse-grained quartz and massive spodumene and montebrasite crystals, with small amounts of K-feldspar. Other accessory minerals are columbite-group minerals, wodginite-group minerals, tapiolite-(Fe), microlite, beryl, cassiterite, pollucite, minjiangite, lazulite, kulanite and strontiohurlbutite. Zone V is the pegmatite core and consists of blocky quartz and K-feldspar crystals and a small amount of accessory minerals. The Nanping No. 31 pegmatite is the type locality of nanpingite [25], strontiohurlbutite [26] and minjiangite [27].

All samples were collected from the mine opening at the 515 m level, and twenty-four mineralized samples from five zones were studied in detail; the occurrence and schematic sequence of rock-forming minerals and Sr-bearing minerals are given in Table 1. The contents of Rb and Sr in K-feldspar from the Nanping No. 31 pegmatite are up to 11,339 and 283 ppm, respectively [28]. Amounts of Sr-bearing phosphates occur in the different zones of the Nanping No. 31 pegmatite.

**Table 1.** List of parts of minerals and their schematic sequence in the Nanping No. 31 pegmatite.

| Chemical Formula | Mineral Name | Magmatic → Hydrothermal | I | II | III | IV | V |
|---|---|---|---|---|---|---|---|
| | | | | | Zone | | |
| $SiO_2$ | Quartz | | ○ | ○ | ○ | ○ | ○ |
| $NaAlSi_3O_8$ | Albite | | √ | √ | √ | ○ | # |
| $KAlSi_3O_8$ | K-feldspar | | | | | ○ | ○ |
| $KAl_2(AlSi_3O_{10})(OH)_2$ | Muscovite | | √ | ○ | ○ | ○ | # |
| $Ca_5(PO_4)_3(F,OH)$ | Apatites | | ○ | ○ | # | # | ○ |
| $LiAl(PO_4)(OH,F)$ | Montebrasite | | # | | ○ | √ | |
| $LiFe(PO_4)$ | Triphylite | | | | # | # | |
| $MgAl_2(PO_4)_2(OH)_2$ | Lazulite | | | | ○ | √ | |
| $SrLi_2Al_4(PO_4)_4(OH)_4$ | Palermoite | | # | | ○ | √ | # |
| $CaLi_2Al_4(PO_4)_4(OH)_4$ | Bertossaite | | | | ○ | √ | |
| $SrAl_3(PO_4)_2(OH)_5 \cdot (H_2O)$ | Goyazite | | | # | # | # | |
| $CaAl_3(PO_4)_2(OH)_5 \cdot (H_2O)$ | Crandallite | | | | | # | |
| $CaBe_2(PO_4)_2$ | Hurlbutite | | # | # | | # | |
| $SrBe_2(PO_4)_2$ | Strontiohurlbutite | | # | # | | # | |
| $BaBe_2(PO_4)_2$ | Minjiangite | | | | | # | |
| $BaNa_2Ca(Fe^{2+},Mn,Mg)_{13}$ $Al(PO_4)_{11}(PO_3OH)(F,OH)_2$ | Fluorarrojadite- (BaNa) | | | | # | # | |

Note: √, ○ and # denote abundant, common and rare, respectively.

### 3. Analytic Methods

#### 3.1. Electron Microprobe Analyses

The chemical compositions of minerals were obtained by using a SHIMADZU 1720H electron microprobe, using wavelength dispersion spectrometry (WDS mode, 15 kV, 20 nA, beam diameter 1 μm), in the School of Earth Sciences, Zhejiang University. Element peaks and backgrounds were measured with counting times of 10 and 10 s, respectively. Standards were used as follows: albite (Si), hornblende (Na, K, Mg, Al, Ca and Fe), fluorapatite (P and F), $MnTiO_4$ (Mn and Ti), synthetic $Ba_3(PO_4)_2$ (Ba), synthetic $SrSO_4$ (Sr) and rubidium titanium phosphate (Rb). A ZAF program was used for all data reduction. Structural formulas were calculated on the basis of 2, 3 and 4 (P + Si) atoms for the hurlbutite group minerals, apatite group minerals and palermoite group minerals, respectively, and 8 O atoms for K-feldspar and albite. The contents of $Li_2O$, BeO and $H_2O$ were estimated by stoichiometric calculation.

#### 3.2. MC-ICP-MS Isotopic Analyses

In situ Sr isotopic analyses were conducted, using multi-collector inductively coupled plasma mass spectrometry (MC-ICP-MS) with 193 nm excimer laser, at State Key Laboratory of Lithospheric Evolution, Institute of Geology and Geophysics, Chinese Academy of Sciences. A spot size of 90 μm was employed with a 6–8 Hz repetition rate and an energy density of 10 $J/cm^2$, depending on the Sr concentration of samples. The Sr isotopic data were acquired by static multi-collection in a low-resolution mode, using nine Faraday collectors. Prior to laser analyses, the Neptune MC-ICP-MS was tuned by using a standard solution to obtain maximum sensitivity. A typical data acquisition cycle consisted of a 40 s measurement of the Kr gas blank with the laser switched off, followed by 60 s of measurement with the laser ablating. Every ten sample analyses were followed by one AP1 apatite reference material measurement for external calibration. Slyudyanka apatite was analyzed in each analytical session and treated as an unknown sample during the data-reduction procedure. The $^{87}Sr/^{86}$ ratios were calculated and normalized from the interference-corrected $^{86}Sr/^{88}$ ratio, using the exponential law. The whole data-reduction procedure was performed by using an in-house Excel VBA (Visual Basic for Applications) macro program.

### 4. Results

#### 4.1. Chemical Compositions of "Primary Sr Minerals"

In granitic systems, plagioclase and/or K-feldspar are the predominant Sr reservoirs, absorb most of the whole-rock Sr and were thus considered as the "primary Sr minerals" in granitic systems [29,30]. In the Nanping No. 31 pegmatite, K-feldspar and albite are the "primary Sr minerals". K-feldspar mainly occurs in zones IV and V, while albite is distributed in the whole pegmatic zones, and secondary albite mainly occurs along the fractures of K-feldspar and spodumene. The chemical compositions of K-feldspar and albite are given in Table 2. K-feldspar in zone IV contains up to 0.05 wt.% SrO, 1.81 wt.% $Rb_2O$, 0.37 wt.% $Na_2O$ and 0.63 wt.% $P_2O_5$, and up to 0.06 wt.% SrO, 1.76wt.% $Rb_2O$, 0.53 wt.% $Na_2O$, 0.26wt.% BaO and 0.39 wt.% $P_2O_5$ in zone V. Overall, the contents of Sr and Rb in albite are lower than those of K-feldspar. The primary albite has up to 0.01 wt.% SrO, 0.09 wt.% $Rb_2O$, 0.07 wt.% $K_2O$ and 0.29 wt.% $P_2O_5$, and secondary albite contains up to 0.10 wt.% SrO, 0.10 wt.% $Rb_2O$, 0.15 wt.% $K_2O$ and 0.40 wt.% $P_2O_5$.

#### 4.2. Petrography and Chemical Compositions of Sr Minerals

##### 4.2.1. Palermoite $Li_2SrAl_4(PO_4)_4(OH)_4$

A lot of Sr phosphates in this study are rare in nature (Table 1). Palermoite was firstly found in the Palermo pegmatite, New Hampshire (USA) [31]. In the Nanping No. 31 pegmatite, palermoite occurs as one of secondary phases of montebrasite in zones III and IV. It is generally distributed along the rims of the primary montebrasite (Figure 2a) or around the primary montebrasite (Figure 2b). In some cases,

palermoite occurs as nodules up to 600 μm in width among the fine-grained muscovite (Figure 2c). Palermoite veinlets, together with lazulite or fine-grained muscovite, are up to several centimeters long and occur in the fractures of primary montebrasite (Figure 2d). These veinlets are generally crosscut by late hydroxylapatite veinlets. It must be noted that palermoite was commonly altered to bertossaite (Figure 2b). Chemically, palermoite contains 41.12–44.87 wt.% $P_2O_5$, 29.79–33.52 wt.% $Al_2O_3$, 8.02–13.74 wt.% SrO, 0.60–3.79 wt.% CaO and 0–1.17 wt.% F and has small amounts of Ba, Fe, Mg, Mn, K, Na and Si (Table 3).

**Table 2.** Representative chemical compositions of K-feldspar and albite in the Nanping No. 31 pegmatite.

| | K-Feldspar | | | | Albite | | | |
|---|---|---|---|---|---|---|---|---|
| | Zone IV | | Zone V | | Primary | | Secondary | |
| $SiO_2$ | 64.08 | 63.62 | 63.75 | 63.00 | 68.70 | 67.31 | 68.52 | 67.93 |
| $Al_2O_3$ | 18.75 | 18.80 | 18.82 | 18.87 | 20.35 | 19.69 | 20.12 | 20.36 |
| $K_2O$ | 16.07 | 16.18 | 16.13 | 15.85 | 0.02 | 0.07 | 0.09 | 0.09 |
| $Na_2O$ | 0.35 | 0.37 | 0.24 | 0.30 | 11.45 | 11.01 | 11.29 | 11.24 |
| CaO | - | - | - | - | 0.05 | 0.02 | 0.02 | 0.02 |
| SrO | 0.02 | 0.05 | 0.06 | 0.04 | 0.01 | 0.01 | 0.10 | - |
| BaO | - | - | 0.20 | 0.26 | - | 0.02 | - | 0.03 |
| $Rb_2O$ | 1.43 | 1.50 | 1.18 | 1.25 | - | - | - | - |
| $P_2O_5$ | 0.15 | 0.06 | 0.10 | 0.15 | 0.08 | 0.04 | - | 0.21 |
| Total | 100.85 | 100.56 | 100.48 | 99.72 | 100.65 | 98.16 | 100.14 | 99.86 |
| | Based on O = 8 | | | | | | | |
| Si | 2.968 | 2.963 | 2.965 | 2.954 | 2.975 | 2.987 | 2.984 | 2.965 |
| Al | 1.024 | 1.032 | 1.032 | 1.043 | 1.039 | 1.030 | 1.033 | 1.047 |
| K | 0.950 | 0.961 | 0.957 | 0.948 | 0.001 | 0.004 | 0.005 | 0.005 |
| Na | 0.031 | 0.033 | 0.021 | 0.028 | 0.961 | 0.947 | 0.953 | 0.951 |
| Ca | | | | | 0.002 | 0.001 | 0.001 | 0.001 |
| Sr | 0.000 | 0.001 | 0.002 | 0.001 | 0.000 | 0.000 | 0.002 | |
| Ba | 0.000 | 0.000 | 0.004 | 0.005 | | 0.000 | | 0.001 |
| Rb | 0.043 | 0.045 | 0.035 | 0.038 | | | | |
| P | 0.006 | 0.002 | 0.004 | 0.006 | 0.003 | 0.002 | | 0.008 |

Note: "-" means below the detection limits.

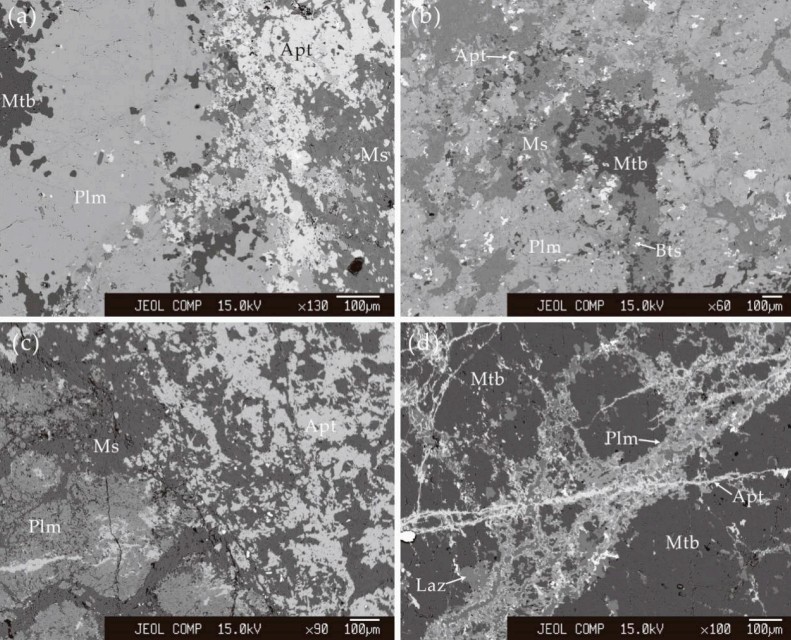

**Figure 2.** Backscattered electron (BSE) images of palermoite assemblages from the Nanping No. 31 pegmatite. (**a**) Palermoite–fluorapatite aggregate around primary montebrasite. (**b**) Palermoite–bertossaite aggregates around primary montebrasite. (**c**) Palermoite nodules among the alteration products of montebrasite. (**d**) Palermoite veinlets in the fractures of montebrasite. Notes: Mtb = montebrasite, Plm = palermoite, Apt = apatites, Ms = muscovite, Bts = bertossaite and Laz = lazulite.

**Table 3.** Representative chemical compositions of palermoite, Sr-rich bertossaite, Sr-rich apatites, Sr-rich hurlbutite and goyazite.

| | Palermoite | | Sr-Rich Bertossaite | | Sr-Rich Apatites | | Sr-Rich Hurlbutite | | Goyazite | |
|---|---|---|---|---|---|---|---|---|---|---|
| $P_2O_5$ | 42.64 | 43.61 | 45.78 | 44.19 | 40.16 | 38.20 | 53.59 | 53.86 | 30.97 | 31.31 |
| $SiO_2$ | - | 0.45 | 0.22 | 0.20 | - | **-** | 0.04 | 0.00 | 0.13 | 0.12 |
| CaO | 0.55 | 3.36 | 5.62 | 4.70 | 47.73 | 43.58 | 15.25 | 13.00 | 2.73 | 0.17 |
| FeO | 0.24 | 0.57 | 0.19 | 0.22 | - | 0.02 | 0.03 | - | 0.02 | 0.17 |
| MnO | 0.23 | 0.17 | 0.04 | 0.07 | 0.27 | 2.71 | 0.01 | - | 0.07 | **-** |
| MgO | 0.14 | 0.22 | 0.08 | 0.11 | - | 0.01 | - | - | 0.02 | - |
| $TiO_2$ | - | 0.03 | 0.01 | 0.14 | - | 0.04 | **-** | **-** | 0.01 | 0.07 |
| SrO | 12.56 | 9.21 | 4.20 | 6.37 | 8.95 | 11.78 | 11.08 | 14.01 | 15.52 | 18.96 |
| BaO | - | 0.20 | 0.09 | - | - | - | 0.01 | 0.05 | 0.87 | 0.28 |
| $Al_2O_3$ | 33.52 | 31.21 | 32.82 | 32.27 | - | 0.01 | 0.10 | 0.04 | 33.98 | 34.00 |
| $Na_2O$ | 0.19 | 0.09 | 0.03 | 0.12 | 0.04 | - | 0.03 | - | 0.06 | 0.01 |
| $K_2O$ | - | - | 0.02 | 0.03 | 0.04 | 0.04 | - | 0.01 | 0.02 | 0.02 |
| F | 0.04 | 0.65 | 0.54 | 0.62 | 0.76 | 1.25 | | | - | 0.10 |
| $H_2O$ | 5.39 | 5.29 | 5.58 | 5.34 | 1.34 | 1.02 | | | 15.35 | 15.46 |
| $Li_2O$ * | 4.49 | 4.64 | 4.84 | 4.68 | | | | | | |
| BeO * | | | | | | | 18.90 | 18.98 | | |
| O = 2F | −0.02 | −0.27 | −0.23 | −0.26 | −0.32 | −0.52 | | | −0.00 | −0.04 |
| | 99.97 | 99.42 | 99.84 | 98.80 | 98.95 | 98.13 | 99.04 | 99.95 | 99.74 | 100.62 |
| *P + Si =* | | | 4 | | 3 | | 2 | | 2 | |
| P | 4.000 | 3.952 | 3.978 | 3.978 | 3.000 | 3.000 | 1.998 | 2.000 | 1.990 | 1.991 |
| Si | | 0.048 | 0.022 | 0.022 | | | 0.002 | 0.000 | 0.010 | 0.009 |
| Ca | 0.064 | 0.381 | 0.611 | 0.529 | 4.455 | 4.276 | 0.711 | 0.603 | 0.219 | 0.013 |
| Fe | 0.022 | 0.051 | 0.017 | 0.020 | | 0.001 | 0.001 | | 0.001 | 0.011 |
| Mn | 0.021 | 0.015 | 0.004 | 0.006 | 0.020 | 0.213 | 0.000 | | 0.004 | |
| Mg | 0.023 | 0.035 | 0.012 | 0.018 | | 0.001 | | | 0.002 | |
| Ti | | 0.002 | 0.001 | 0.011 | | 0.002 | | | 0.001 | 0.004 |
| Sr | 0.807 | 0.572 | 0.250 | 0.393 | 0.458 | 0.633 | 0.283 | 0.356 | 0.683 | 0.826 |
| Ba | | 0.008 | 0.004 | | | | 0.000 | 0.001 | 0.026 | 0.008 |
| Al | 4.377 | 3.937 | 3.970 | 4.045 | 0.000 | 0.001 | 0.005 | 0.002 | 3.039 | 3.010 |
| Na | 0.042 | 0.018 | 0.007 | 0.025 | 0.006 | | 0.002 | | 0.009 | 0.001 |
| K | | | 0.003 | 0.004 | 0.004 | 0.005 | | 0.001 | 0.002 | 0.002 |
| Li | 2.000 | 2.000 | 2.000 | 2.000 | | | | | | |
| Be | | | | | | | 2.000 | 2.000 | | |
| F | 0.014 | 0.219 | 0.174 | 0.207 | 0.212 | 0.366 | | | | 0.023 |
| OH | 3.986 | 3.781 | 3.826 | 3.793 | 0.788 | 0.634 | | | 5.000 | 4.977 |

Notes: FeO as the total Fe; "-" means below the detection limits; *: calculated by stoichiometry.

### 4.2.2. Goyazite $SrAl_3(PO_4)_2(OH)_5 \cdot H_2O$

Goyazite occurs as secondary phase in the Nanping No. 31 pegmatite; it generally formed veinlets hundreds of μm in length, distributed along the fractures of albite in zone II (Figure 3a) and the fractures of bertossaite in zone IV (Figure 3b). The association minerals are hydroxylapatite, kulanite, secondary montebrasite and fine-grained muscovite. Electron microprobe analyses (Table 3) show that goyazite contains 30.06–34.13 wt.% $P_2O_5$, 32.90–36.03 wt.% $Al_2O_3$, 13.92–20.19 wt.% SrO, 0.09–3.55 wt.% CaO, 0.28–3.51 wt.% BaO and 0–2.24 wt.% F and has minor amounts of Mg, Fe, Mn, Ti, Na, K and Si.

### 4.2.3. Strontiohurlbutite $SrBe_2(PO_4)_2$

Strontiohurlbutite is a Sr-Be phosphate found in the Nanping No. 31 pegmatite [26]. It occurs as euhedral to subhedral crystals in zone I, as aggregates in Be silicate + phosphate associations in zone II, and as one of the alteration products of hurlbutite (zone I) and beryl (zone IV). Some crystals occur among quartz crystals in zone I (Figure 4a), or together with palermoite, apatites and muscovite are distributed along the fractures of primary montebrasite in zone IV (Figure 4b). Chemically,

strontiohurlbutite contains 50.54–54.15 wt.% $P_2O_5$, 20.96–30.61 wt.% SrO and 0.07–6.97 wt.% CaO and has a small amount of BaO.

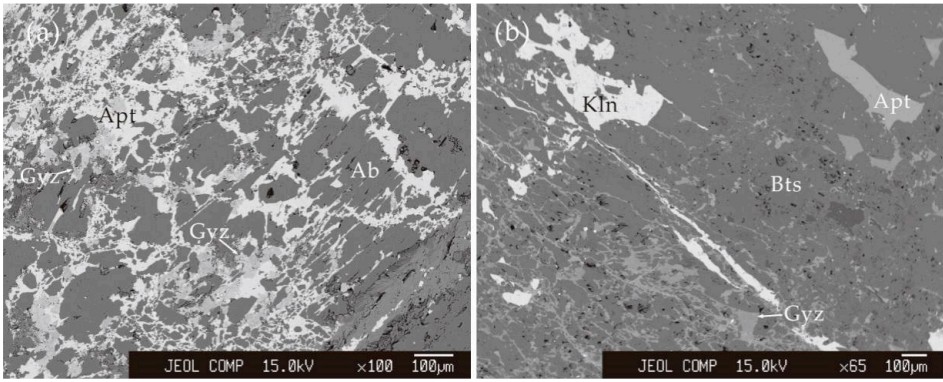

**Figure 3.** BSE images of goyazite assemblages from the Nanping No. 31 pegmatite. (**a**) Goyazite with hydroxylapatite along the fractures of albite from the zone II. (**b**) Goyazite veinlets in the fractures of bertossaite. Notes: Gyz = goyazite, Apt = apatites, Ab = albite, Bts = bertossaite and Kln = kulanite.

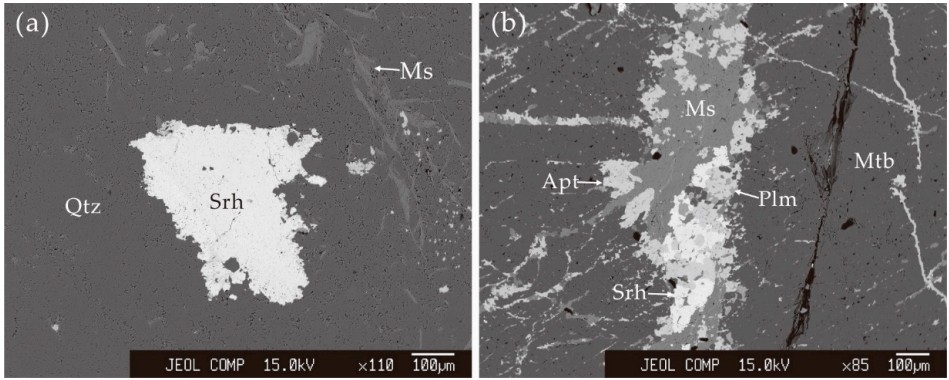

**Figure 4.** BSE images of strontiohurlbutite assemblages from the Nanping No. 31 pegmatite. (**a**) Strontiohurlbutite crystal among quartz. (**b**) Strontiohurlbutite together with palermoite and apatites along the fractures of montebrasite. Notes: Srh = strontiohurlbutite, Qtz = quartz, Ms = muscovite, Apt = apatites, Plm = palermoite and Mtb = montebrasite.

*4.3. Petrography and Chemical Compositions of Sr-Rich Minerals*

4.3.1. Apatites $Ca_5(PO_4)_3(F,OH)$

Apatites (apatite group minerals) in the Nanping No. 31 pegmatite dyke occur as primary and secondary phases in different zones (Table 1). Primary apatites formed subhedral to enhedral crystals up to 20 centimeters in length and are closely associated with rock-forming minerals, such as quartz and albite (Figure 5a,b). A portion of apatite crystals were altered but keep their original form, and remnants of apatite were found among quartz crystals (Figure 5a). Secondary apatites mainly occur as the alteration products of primary and early phosphate minerals in different zones. Large quantities of secondary apatite, however, are enriched in Sr among the associations of montebrasite alteration. Four typical different modes of secondary Sr-rich apatites are distinguished according to the petrological features and chemical compositions. The first occurrence is characterized by subhedral to euhedral Sr-rich apatite crystals, which are replaced by late Sr-poor hydroxylapatite along their rims with remnants of Sr-rich apatite in the center (Figure 5c). The most common occurrence of Sr-rich apatites is represented by irregular aggregates/veinlets, intergrown with secondary quartz and fine-grained muscovite in the fractures of or surrounding montebrasite. These veinlets generally replaced the early

secondary hydroxylapatite and palermoite along their fractures and rims (Figure 5d). Mosaic-like Sr-rich apatites in fine grained muscovite assemblages of montebrasite are the third occurrence of Sr-rich apatites (Figure 5e). On the backscattered electron image, the mosaic consists of three distinct domains: a brighter, transitional and darker area. The brighter area is sporadically distributed in the transitional area and has higher contents of SrO (approximately 11.78 wt.%); the transitional area has 5.82 wt.% SrO, and the darker area is relatively poor in Sr (only 0.33 wt.% SrO). Sr-rich apatite rims up to 5 µm in width are the fourth occurrence of Sr-rich apatite. They generally surround the Sr-poor apatite (Figure 5f).

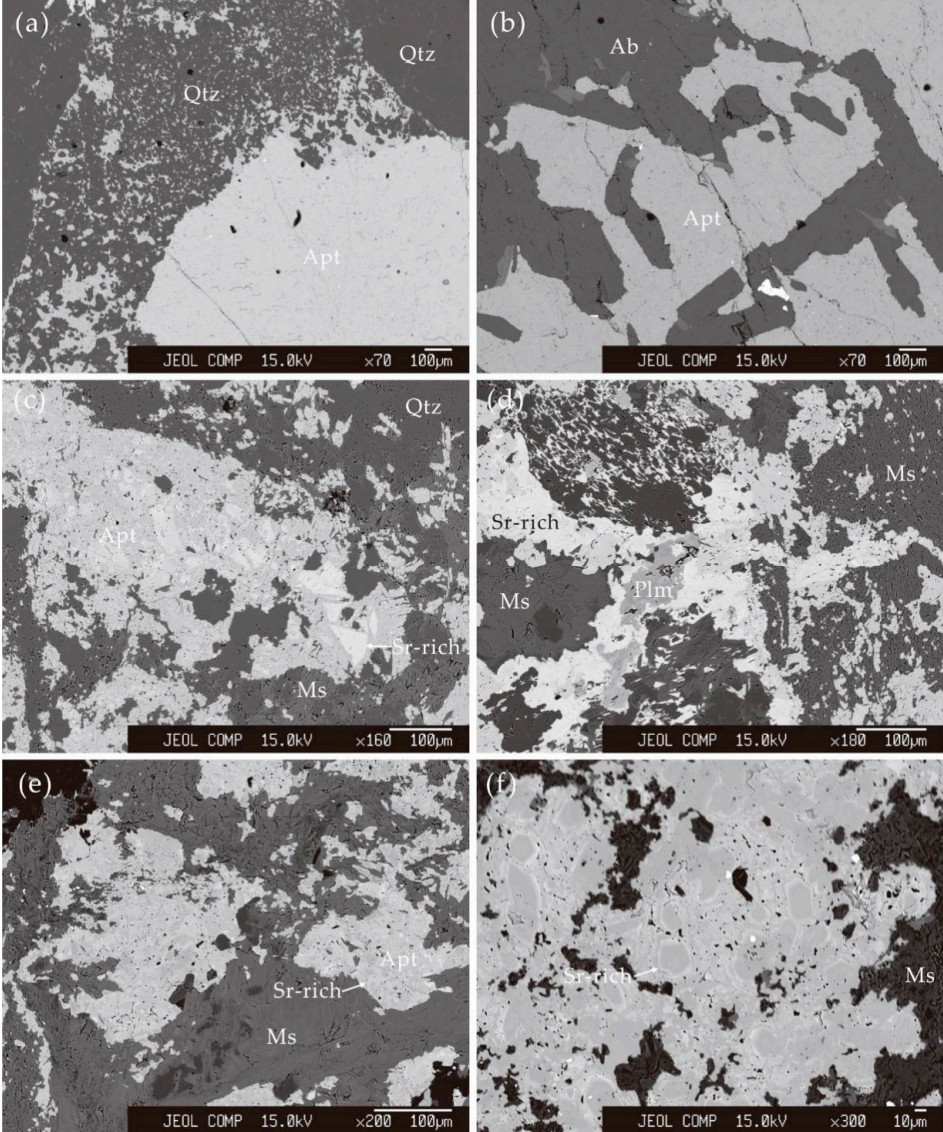

**Figure 5.** BSE images of apatites association from the Nanping No. 31 pegmatite dyke. (**a**) Primary apatites among quartz. (**b**) Primary apatites associated with albite. (**c**) Secondary apatites with Sr-rich core and Sr-poor rims. (**d**) Irregular Sr-rich apatite aggregates/veinlets replaced hydroxylapatite and palermoite among fine-grained muscovite among quartz. (**e**) Mosaic-like Sr-rich apatites interstitial to fine grained muscovite. (**f**) Sr-rich fluorapatite rims among fine-grained muscovite. Notes: Qtz = quartz, Apt = apatites, Ab = albite, Ms = muscovite and Plm = palermoite.

Electron microprobe analyses indicate that only Sr-rich apatite rims with an average content of 3.22 wt.% F, corresponding to fluorapatite, while the others are hydroxylapatites (Table 3). These apatites

contain 1.13 to 11.78 wt.% SrO, with a slight enrichment in Mn (up to 6.65 wt.% MnO), as well as small amounts of BaO (<0.57 wt.%), FeO (<3.03 wt.%), MgO (<0.17 wt.%), $SiO_2$ (<2.35 wt.%), $Al_2O_3$ (<0.99 wt.%), $TiO_2$ (<0.51 wt.%), $Na_2O$ (<0.34 wt.%) and $K_2O$ (<0.97 wt.%). The chemical compositions of primary apatites in the Nanping No. 31 pegmatite were given by Reference [32]. On the whole, secondary apatites are more enriched in Sr than primary apatites; the evolution of SrO and MnO/FeO in secondary apatites is described in our literature (Figure 5; [32]).

### 4.3.2. Hurlbutite $CaBe_2(PO_4)_2$

Hurlbutite occurs as discrete crystals, aggregates and as one of the alteration products of montebrasite and beryl in the different internal zones. Discrete crystals are euhedral and approximately 300 μm in width and included in quartz from zone I. The crystals were generally replaced by late hydroxylapatite and strontiohurlbutite. Hurlbutite aggregates mainly occur in the Be silicate + phosphate assemblages in zone II; they are closely associated with beryl, phenakite, fluorapatite, hydroxylherderite and strontiohurlbutite, as well as interstices of albite. Hurlbutite was found as one of the alteration products of montebrasite in zone IV; it formed subhedral to euhedral crystals, ranging from 50 to 200 μm in length in interstices of fine-grained muscovite; the associated minerals include secondary wagnerite and hydroxylapatite. Hurlbutite also occurs as one of the alteration products of Cs-rich beryl in zone IV, associated with secondary strontiohurlbutite, muscovite and hydroxylapatite. Electron microprobe results show a high enrichment of Sr in hurlbutite and different contents of SrO in different types of hurlbutite (Table 3). The discrete hurlbutite crystals from zone I contain up to 12.03 wt.% SrO, hurlbutite aggregates from zone II up to 11.08 wt.% SrO, hurlbutite as an alteration product of montebrasite from zone IV < 7.55 wt.% SrO and hurlbutite veinlets of alteration products from beryl in zone IV up to 14.01 wt.% SrO.

### 4.3.3. Bertossaite $Li_2CaAl_4(PO_4)_4(OH)_4$

Bertossaite is one of the end-members of a Sr–Ca isomorphic series with an ideal chemical formula of $Li_2(Ca,Sr)Al_4(PO_4)_4(OH)_4$. It generally occurs as an alteration product of montebrasite in the Nanping No. 31 pegmatite, forming relatively earlier than palermoite. However, in some cases, it also occurs along the fractures of, or surrounding, palermoite grains (Figure 3b). Chemically, bertossaite is enriched in Sr and contains up to 6.37 wt.% SrO (Table 3).

### 4.3.4. Fluorarrojadite-(BaNa) $BaNa_2Ca(Fe^{2+},Mn,Mg)_{13}Al(PO_4)_{11}(PO_3OH)(F,OH)_2$

Fluorarrojadite-(BaNa) was found as anhedral grains, sporadically distributed along rims and/or fractures of lazulite in zone IV. The associated minerals include secondary montebrasite, fluorapatite and palermoite. This mineral was also found as one of the alteration products of triphylite from zones III and IV. Chemical composition data show that arrojadite-group minerals are not only enriched in Ba, Na and F, and indicative of fluorarrojadite-(BaNa), but also have a high concentration of Sr in both types of fluorarrojadite-(BaNa). They have 0.28–4.13 wt.% SrO, with 41.43–43.24 wt.% $P_2O_5$, 2.48–2.60 wt.% $Al_2O_3$, 5.56–6.70 wt.% $Na_2O$, 23.91–25.66 wt.% FeO, 6.68–9.91 wt.% MnO, 6.40–9.10 wt.% MgO, 0.59–7.22 wt.% BaO, 1.67–2.40 wt.% CaO and 0.99–1.32 wt.% F.

### 4.3.5. Crandallite $CaAl_3(PO_4)_2(OH)_5 \cdot H_2O$

Crandallite was only detected in one thin-section. This mineral forms veinlets up to several millimeters in length, within the fractures of the primary montebrasite in zone IV. Strontiohurlbutite and fluorapapatite were observed in these veinlets. Chemically, crandallite is enriched in Sr and contains 3.50 wt.% SrO, as well as 32.07 wt.% $P_2O_5$, 35.15 wt.% $Al_2O_3$ and 11.93 wt.% CaO.

## 4.4. Isotopic Features of $^{87}Sr/^{86}$

The Sr isotopic results of Sr-bearing phosphates obtained by MC-ICP-MS show different isotopic features of $^{87}Sr/^{86}$ (Table 4 and Figure 6). Primary apatites have an average $^{87}Sr/^{86}$ ratio of 0.72625, but the $^{87}Sr/^{86}$ ratios of secondary apatites range from 0.72747 to 0.73357, with an average of 0.72997. Two crystals of strontiohurbutite were analyzed, showing an average $^{87}Sr/^{86}$ ratio of 0.73026. The average $^{87}Sr/^{86}$ ratios of bertossaite and palermoite are 0.72618 and 0.73408, respectively. According to the U-Pb age of 387 Ma from the columbite-(Fe) and zircon of 387 Ma in the Nanping No. 31 pegmatite [33], the calculated average $(^{87}Sr/^{86})_i$ according to the decay rate of Rb [34] yields a wide range of $(^{87}Sr/^{86})_i$ in Sr minerals and Sr-rich minerals. The $(^{87}Sr/^{86})_i$ of the primary apatites range from 0.72598 to 0.72669, 0.72744 to 0.73357 in secondary apatites, 0.72988 to 0.73058 in strontiohurbutite, 0.72543 to 0.72849 in bertossaite and 0.73309 to 0.73448 in palermoite, respectively.

**Table 4.** The MC-ICP-MS Sr isotopic analytical results of Sr-bearing mineral from the Nanping No. 31 pegmatite.

| Sample | $^{84}Sr/^{86}$ (±2SD) | $^{84}Sr/^{88}$ (±2SD) | $^{87}Rb/^{86}$ (±2SD) | $^{87}Sr/^{86}$ (±2SD) | $(^{87}Sr/^{86})_i$ |
|--------|--------|--------|--------|--------|--------|
| P-Apt | 0.0563(6) | 0.00673(7) | 0.0007(1) | 0.72598(10) | 0.72598 |
| P-Apt | 0.0565(9) | 0.00674(11) | 0.0002(0) | 0.72669(24) | 0.72669 |
| P-Apt | 0.0567(8) | 0.00677(9) | 0.0015(1) | 0.72625(16) | 0.72624 |
| P-Apt | 0.0569(13) | 0.00679(16) | 0.0005(0) | 0.72608(14) | 0.72608 |
| S-Apt | 0.05640(1) | 0.00674(2) | 0.0062(7) | 0.72854(4) | 0.72851 |
| S-Apt | 0.05667(3) | 0.00677(4) | 0.0264(33) | 0.73224(7) | 0.73210 |
| S-Apt | 0.0563(11) | 0.00672(13) | 0.0012(1) | 0.73157(16) | 0.73156 |
| S-Apt | 0.0565(9) | 0.00674(10) | 0.0064(6) | 0.72747(15) | 0.72744 |
| S-Apt | 0.0561(11) | 0.00670(13) | 0.0020(2) | 0.72849(14) | 0.72848 |
| S-Apt | 0.0567(7) | 0.00677(9) | 0.0017(1) | 0.72900(14) | 0.72899 |
| S-Apt | 0.0564(6) | 0.00673(8) | 0.0002(0) | 0.72806(12) | 0.72806 |
| S-Apt | 0.0566(1) | 0.00675(1) | 0.0000(0) | 0.73357(4) | 0.73357 |
| S-Apt | 0.0565(1) | 0.00675(1) | 0.0000(0) | 0.72954(4) | 0.72954 |
| S-Apt | 0.0564(1) | 0.00674(1) | 0.0000(0) | 0.72990(3) | 0.72990 |
| S-Apt | 0.0565(1) | 0.00675(2) | 0.0000(0) | 0.73366(5) | 0.73366 |
| S-Apt | 0.0565(1) | 0.00674(1) | 0.0000(0) | 0.72869(5) | 0.72869 |
| S-Apt | 0.0565(0) | 0.00675(1) | 0.0000(0) | 0.72891(2) | 0.72891 |
| Srh | 0.0565(0) | 0.0067(1) | 0.0046(11) | 0.73060(3) | 0.73058 |
| Srh | 0.0565(1) | 0.00675(1) | 0.0071(4) | 0.72992(5) | 0.72988 |
| Bts | 0.0560(5) | 0.00668(5) | 0.0001(0) | 0.72552(9) | 0.72552 |
| Bts | 0.0563(3) | 0.00672(3) | 0.0000(0) | 0.72605(6) | 0.72605 |
| Bts | 0.0564(2) | 0.00673(3) | 0.0000(0) | 0.72594(6) | 0.72594 |
| Bts | 0.0559(3) | 0.00668(4) | 0.0001(0) | 0.72543(7) | 0.72543 |
| Bts | 0.0563(5) | 0.00672(6) | 0.0000(0) | 0.72564(12) | 0.72564 |
| Bts | 0.0564(1) | 0.00674(1) | 0.0000(0) | 0.72849(3) | 0.72849 |
| Plm | 0.0562(9) | 0.00671(11) | 0.0212(6) | 0.73320(20) | 0.73309 |
| Plm | 0.0564(11) | 0.00674(13) | 0.0490(26) | 0.73415(21) | 0.73389 |
| Plm | 0.0564(12) | 0.00674(14) | 0.0743(20) | 0.73488(26) | 0.73448 |

Notes: P-Apt = primary apatite; S-Apt = secondary apatite; Srh = strontiohurbutite; Bts = bertossaite; Plm = palermoite; SD = standard deviation.

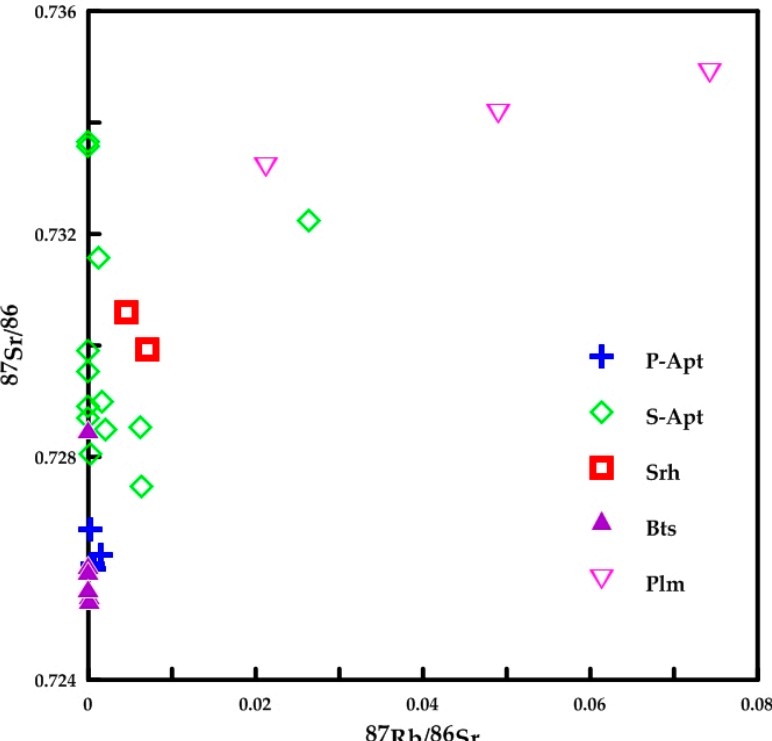

**Figure 6.** Rb–Sr isochron diagram of Sr minerals and Sr-rich minerals form the Nanping No. 31 pegmatite.
Notes: P-Apt = primary apatite; S-Apt = secondary apatite; Srh = strontiohurbutite; Bts = bertossaite;
Plm = palermoite.

## 5. Discussion

### 5.1. Crystal–Chemical Features of Strontium in Pegmatic Systems

Strontium is one of lithophile elements, and the ionic radius of $Sr^{2+}$ is 1.26 Å (8-fold coordination)
and 1.44 Å (12-fold coordination) between $Ca^{2+}$ and $K^+$, but similar to $Na^+$ [35,36]. Therefore, strontium
is compatible in most rock-forming minerals, dispersed as one trace element in Ca-bearing minerals
and K-bearing minerals [1]. In pegmatites, Sr mainly entered into the crystal structures of alkali
feldspar, and the partition coefficients of Sr in alkali feldspar showed positive correlations with its
$Al_2O_3$ [37]. Thus, the substitution of $Sr^{2+} + Al^{3+} \leftrightarrow K^+/(Na^+) + Si^{4+}$ probably occurs in alkali feldspar,
where Sr substitutes for K, accompanied by the substitution of Si by Al. In the Nanping No. 31
pegmatite, K-feldspar contains up to 0.05 wt.% SrO in zone IV and 0.06 wt.% SrO in zone V; primary
and secondary albite contain up to 0.01 wt.% SrO and 0.10 wt.% SrO, respectively (Table 2). These
features indicate the substitutions of Sr for K and Na in crystal structures of alkali feldspar.

Moreover, Sr is geochemically similar to the other alkali earth elements Ca and Ba, and it is thus
easily incorporated in Ca-bearing minerals and Ba-bearing minerals and resulted in the substitutions
of $Sr^{2+} \leftrightarrow Ca^{2+}$ and $Sr^{2+} \leftrightarrow Ba^{2+}$. In the Nanping No. 31 pegmatite, the substitution of $Sr^{2+} \leftrightarrow Ca^{2+}$
was demonstrated in secondary apatite, hurlbutite and bertossaite [32]. The contents of SrO is up to
11.78 wt.% in secondary apatites, 12.03 wt.% in hurlbutite and 7.98 wt.% in bertossaite, respectively.
However, $Sr^{2+}$ ions occupied in different coordinated polyhedra among the crystal-structures of
these phosphates, such as 6-fold coordinated polyhedra in apatites, 10-fold coordinated polyhedra in
hurlbutite and 8-fold coordinated polyhedra in bertossaite, respectively. The equivalent substitution
of Ba by Sr generally occurs in the crystal structures of Ba-bearing phosphates. Kulanite and
minjiangite from the Nanping No. 31 pegmatite have up to 2.50 and 0.24 wt.% SrO, respectively.
Fluorarrojadite-(BaNa) contains up to 4.13 wt.% SrO, also indicating a substitution of Ba by Sr in the
crystal structure of the arrojadite-group minerals.

*5.2. Geochemical Sr Recirculation in the Nanping No. 31 Pegmatite*

During magmatic stages of granitic pegmatites, Sr primarily behaves as one of dispersed elements and prefers to partition into the crystal structures of alkali feldspar [38–41]; alkali feldspars were thus considered "primary Sr minerals" in granitic pegmatites, and small amounts of Sr entered primary apatite. However, hydrothermal alteration of granitic pegmatites results in the concentration of Sr [3,15]. The increasing alkalinity of deuteric fluids may facilitate Sr to substitute Ca in apatites, while Sr-bearing minerals crystallized under decreasing or low alkalinity fluids [42]. Both potassic alkaline and sodic alkaline fluids may induce the substitution of Sr for Ca in secondary minerals, but potassic alkaline pegmatites are significantly enriched in Sr relative to the sodic systems [6,43]. Therefore, the alkalinity of hydrothermal fluids is the principal control of the concentration and crystallization of Sr in granitic pegmatites.

In the Nanping No. 31 pegmatite, K-feldspar occurs as the "primary Sr mineral" and contains up to 0.06 wt.% SrO; both primary and secondary albite crystals have up to 0.01 wt.% SrO (Table 2). Thus, the albitization and other hydrothermal alteration of K-feldspar resulted in the concentration of Sr in hydrothermal fluids. The hydrothermal alteration of primary apatites (Figure 5a), which contain up to 0.26 wt.% SrO [32], also contributed small amounts of Sr to hydrothermal fluids. However, the crystallization of Sr-bearing minerals is mainly related to the alkalinity of hydrothermal fluids, which was affected by the alterations of alkaline minerals such as K-feldspar, spodumene, triphylite and montebrasite, etc. Discrete strontiohurlbutite and Sr-rich hurlbutite crystals from zone I are intimately intergrown with quartz and muscovite (Figure 3a), indicating that their formation was linked to the K-stage of hydrothermal alteration. In zones II and III, the assemblages of hurlbutite and strontiohurlbutite typically occur interstitial to primary albite, and goyazite formed as veinlets distributed along the fractures of albite (Figure 3a), reflecting a sodic condition. Fluorarrojadite-(BaNa) grains are enriched in Sr and occur as one of the alteration products of triphylite in zones III and IV [24], suggesting that significant amounts of Na and Sr were sequestered in arrojadite group minerals at the stage of decreasing alkalinity. The crystallization of palermoite associated with fine-grained muscovite during the alteration of montebrasite demonstrated K- and Li-rich conditions. The mosaic-like hydroxylapatites with different SrO contents (Figure 5e) reflect the variation in alkalinity of hydrothermal fluids during the crystallization of apatites. The veinlets of palermoite and goyazite (Figure 3b) were related to the direct precipitation from Sr-rich fluids at a low temperature (about 300 °C). Similar veinlets of goyazite were observed in the San Elías pegmatite [13,14,44].

However, the alkalinity of hydrothermal fluids in the Nanping No. 31 pegmatite may also be affected by externally derived fluids [45,46]. Amounts of external Sr from wall-rocks (schists and granulites) are possibly brought into the pegmatite in an open system [3,12]. The isotopic features of $^{87}Sr/^{86}$ system from Sr-bearing phosphates (Table 4 and Figure 6) show a wide range in $^{87}Sr/^{86}$ with respect to their deviation from isochron; therefore, the Nanping No. 31 pegmatite likely formed under an open system or significant chemical disturbance [47]. The average $^{87}Sr/^{86}$ ratio of bertossaite is 0.72618; some bertossaite crystals were formed by the replacements of palermoite (Figure 2b). These features suggest less integrated radiogenic $^{87}Sr$ during their crystallization. Strontiohurbutite with relative higher $^{87}Sr/^{86}$ ratio (Table 4 and Figure 6) likely formed under moderately integrated radiogenic $^{87}Sr$ fluids. Palermoite occurs as one of secondary phases and generally forms at low temperatures in granitic rocks [3]; it has higher $^{87}Sr/^{86}$ ratios (0.73309–0.73448) than other Sr-bearing phosphates, indicative of more integrated radiogenic $^{87}Sr$ during its crystallization. The $^{87}Sr/^{86}$ ratios of secondary apatites range from 0.72747 to 0.73357, suggesting that secondary apatites crystallized from fluids with different radiogenic signatures through the whole hydrothermal stages of the pegmatite. Therefore, the occurrence, chemical composition and isotopic $^{87}Sr/^{86}$ features of Sr-bearing minerals reflect a significant chemical disturbance of Sr system by externally derived fluids in an open system. Two post-magmatic recirculation processes of Sr are proposed in the Nanping No. 31 pegmatite: (1) The breakdown processes of K-feldspar and primary apatites (Figure 5a) resulted in the concentration of Sr in the hydrothermal fluids and the crystallization of secondary Sr-bearing phosphates, such as

strontiohurlbutite, palermoite, Sr-rich apatites, hurlbutite and bertossaite; and (2) the replacements of palermoite, Sr-rich apatites and hurlbutite by later fluids released Sr into hydrothermal fluids again, which caused the Sr-rich rims of hydroxylapatite (Figure 5f) and direct precipitation of later palermoite and goyazite from later Sr-rich fluids at low temperatures (Figure 3). External fluids from wall-rocks introduced some Sr into the Nanping No. 31 pegmatite.

*5.3. Source of Sr in the Nanping No. 31 Pegmatite*

In granitic pegmatites, internally derived Sr normally provides a Sr source for secondary Sr-bearing phases in a relatively closed system [48]. However, the disturbance of Sr isotope features from Sr-bearing phosphates (Table 4 and Figure 6) suggests multiple Sr sources in the Nanping pegmatite. The Nanping No. 31 pegmatite is one of the abyssal-type pegmatites, formed by anatexis in the lower crust [20]. Most of Sr comes from the hydrothermal alteration of "primary Sr minerals" such as primary K-feldspar and apatites. However, field observations show metasomatism between the pegmatite and its wall-rocks (schists and granulites), and the isotopic study of oxygen ($\delta^{18}O > 9.5‰$) illustrates that external fluids were introduced into the Nanping No. 31 pegmatite [20]. Its wall-rocks (schists and granulites), as far as to ground water, may possibly introduce amounts of Sr into the Nanping pegmatitic system. The isotopic features of $^{87}Sr/^{86}$ from Sr-bearing phosphates in the Nanping No. 31 pegmatite show the wide range of $^{87}Sr/^{86}$ (Table 4 and Figure 6), which demonstrates the participation of the external source of Sr. Therefore, the Nanping No. 31 pegmatite was formed under a relatively open system, and during its crystallization differentiation, external fluids from wall-rocks introduced some Sr into the pegmatitic system.

However, radiogenic $^{87}Sr$ by decay of Rb in feldspars may be another source for secondary Sr-bearing phases [12,49], and Sr from K-feldspar is not inversely correlated with Rb [3]. In the Nanping No. 31 pegmatite, both Sr and Rb in K-feldspar increase from the early stage to the late stage [28], and secondary albite crystals keep lower amounts of Rb during the breakdown of K-feldspar (Table 4). The wide range of $^{87}Rb/^{86}Sr$ and $^{87}Sr/^{86}Sr$ ratios from secondary strontiohurlbutite, bertossaite and palermoite (Table 4 and Figure 6) illustrate that different proportions of radiogenic $^{87}Sr$ participate in crystallization of secondary Sr-bearing minerals. Therefore, Sr in secondary Sr-bearing phosphates from the Nanping No. 31 pegmatite originated from the crystallization differentiation of pegmatite itself, and partly from the external fluids of wall-rock and radiogenic $^{87}Sr$ from Rb.

## 6. Conclusions

(1) In the Nanping No. 31 pegmatite, K-feldspar and primary apatites are the major "primary Sr minerals", the occurrences of secondary Sr phosphates (strontiohurlbutite, palermoite and goyazite) and Sr-rich phosphates (apatites, hurlbutite, bertossaite and fluorarrojadite-(BaNa)) reflect the transport, concentration and recrystallization of Sr in granitic systems.

(2) The mobilization and recrystallization of Sr in granitic systems are principally controlled by the variation in alkalinity of hydrothermal fluids. Increasing alkalinity of fluids facilitates Sr into secondary Ca-bearing minerals to form Sr-rich minerals, such as apatites, hurlbutite, bertossaite and fluorarrojadite-(BaNa). Sharply decreasing alkalinity of fluids by the precipitation of secondary muscovite and albite induces the crystallization of strontiohurlbutite, palermoite and goyazite, respectively. The alkalinity of hydrothermal fluids was buffered by the decomposition of alkaline primary minerals (K-feldspar, spodumene triphylite and montebrasite), which reveals complex and parallel controls on the geochemical mobility of Sr in granitic systems.

(3) Two post-magmatic recirculations of Sr are proposed in the Nanping No. 31 pegmatite: (a) the breakdown of "primary Sr minerals" (K-feldspar and apatites) and crystallization of secondary Sr-bearing phases, such as strontiohurlbutite, palermoite, Sr-rich apatites, hurlbutite and bertossaite; and (b) the replacement of secondary Sr-bearing minerals and direct precipitation of later palermoite and goyazite from later Sr-rich fluids at low temperatures.

(4) The disturbed isotopic features of $^{87}Sr/^{86}$ from Sr-bearing phosphates in the Nanping pegmatite indicate an open system behavior for Sr system. Externally derived fluids introduced amounts of Sr into the pegmatic systems during the emplacement and consolidation of the pegmatite.

**Author Contributions:** Conceptualization, C.R. and R.-C.W.; methodology, R.-C.W. and F.H.; article writing and figure drawing, C.R.; project administration, C.R.; supervision, R.-C.W. and F.H.; sample collection, R.-Q.W. and Q.W.; funding acquisition, C.R. All authors have read and agreed to the published version of the manuscript.

**Funding:** This study was financially supported by the NSF of China (Grant Nos. 41472036 and 41772031).

**Conflicts of Interest:** The authors declare no conflict of interest.

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
