# Peer review of "Mineralogy and Geochemistry of Sr-Bearing Phosphates from the Nanping No. 31 Pegmatite (SE China): Implications for Sr Circulation and Post-Magmatic Processes in Granitic Systems"

_minerals, doi:10.3390/min10060541_

Round 1

Reviewer 1 Report

This paper presents an interesting paragenetic study of Sr phosphates from a pegmatite in China to reveal the complexity of Sr mobility in granitic systems. Whilst the paragenetic and mineral compositional data of these complex intergrowhs of primary and secondary Sr-phosphates is detailed and of general interest, data presentation and analysis is inadequate. The Sr isotope signatures of the minerals are not analysed in context of their Rb/Sr ratios. The discussion focusses on the behaviour of Sr, but needs to also consider Rb contents in the various mineral phases as this is essential for interpreting the Sr isotope signatures. The authors propose that the pegmatite formed by crustal anataxis followed by alteration by wall-rock derived external fluids, without considering the isotopic signatures of those reservoirs. In other words, the conclusions are not supported by the data.

On the basis of my comments, I suggest the manuscript is thoroughly revised before being reconsidered for publication. I hope my comments provide some useful guidelines to improve this work

  1. Mineral compositional data, i.e. Table 2: these show representative data, but how many were analysed? What is the compositional variability of all these minerals? Are there solid solution trends?
  2. A paragenetic diagram of the different mineral phases (mineral versus time/process) would be very helpful.
  3. Analyses & discussion of Sr isotopic data: Please include plots showing the isotopic signatures of different minerals (i.e. isochron plots showing 87Sr/86Sr versus 87Rb/86Sr) and analyse the data in context of their contemporary Rb/Sr ratios. The discussion of Sr isotope data does not make sense without simultaneously discussing mineral Rb/Sr ratios. Should Table 3 not include absolute Rb and Sr concentrations?

I noticed some apatites have 87Rb/86Sr of zero, which would mean they contain no Rb. Is that correct? Are they below detection limit? If true, their Sr isotopic values should theoretically represent initial values because there would have been no radiogenic ingrowth since crystallisation (of course, providing they remained isotopically closed). However, there’s a wide range in their 87Sr/86Sr ratios. So please discuss the nature of this disturbance/variation, i.e. loss of Sr, addition of Rb/Sr, different sources, etc etc.

The data clearly do not follow any kind of isochron trend, again indicating extensive disturbance or geological variation, i.e. different fluid sources and open system behaviour etc. These are things that need to be evaluated properly.

  1. Please discuss Rb/Sr signatures and Sr isotopic data available in the literature on the crustal rocks from which these pegmatites were sourced and into which they were emplaced, in order to support proposed processes of anataxis/metasomatism by external fluids.
  2. What decay rate of Rb is used to calculate initial Sr ratios? I don't see it cited anywhere. Please use the most recent and accurate value of λ87Rb 1.397*10−11 by Villa et al 2015. Mention in data Table and/or methods.
  3. Finally, please have the manuscript proof-read for typos and correct syntax prior to resubmitting. I attached an annotated pdf which I commented on while reading. Please excuse some repetition of comments above. 

I wish the authors the best of luck with this work.

May 2020

Author Response

1.Mineral compositional data, i.e. Table 2: these show representative data, but how many were analysed? What is the compositional variability of all these minerals? Are there solid solution trends?

Response 1: we measured analysed near two hundreds dots, it is difficult to show all data here, but we have described the compositional variability for Sr/Sr-rich minerals. We are sure that there are not solid solution trends between palermoite andbertossaite, hurlbutite and strontiohurlbutite

2.A paragenetic diagram of the different mineral phases (mineral versus time/process) would be very helpful.

Response 2: good suggestion, we add another table to show main mineral versus process (relative time)

3.Analyses & discussion of Sr isotopic data: Please include plots showing the isotopic signatures of different minerals (i.e. isochron plots showing 87Sr/86Sr versus 87Rb/86Sr) and analyse the data in context of their contemporary Rb/Sr ratios. The discussion of Sr isotope data does not make sense without simultaneously discussing mineral Rb/Sr ratios. Should Table 3 not include absolute Rb and Sr concentrations?

Response 3: we have add another Figure to show the 87Sr/86Sr versus 87Rb/86Sr plots for Sr and Sr-rich minerals. And we also discuss the 87Rb/86Sr and radiogenic 87Sr. In Table 3, our data come from the analyses by MC-ICP-MS, it only get the ratios between 87Sr, 86Sr and 87Rb etc. So it is difficult to get to absolute Rb and Sr contents by this methods. We also want to analyse the contents of Rb and Sr in Sr/Sr-rich minerals, but most Sr minerals are secondary, and the sizes of grains or veinlets are difficult to get good data.

4.I noticed some apatites have 87Rb/86Sr of zero, which would mean they contain no Rb. Is that correct? Are they below detection limit? If true, their Sr isotopic values should theoretically represent initial values because there would have been no radiogenic ingrowth since crystallisation (of course, providing they remained isotopically closed). However, there’s a wide range in their 87Sr/86Sr ratios. So please discuss the nature of this disturbance/variation, i.e. loss of Sr, addition of Rb/Sr, different sources, etc etc.

Response 4: some apatites contain Rb, some have a small amounts of Rb, so it is an open system with disturbance by external fluids.

5.The data clearly do not follow any kind of isochron trend, again indicating extensive disturbance or geological variation, i.e. different fluid sources and open system behaviour etc. These are things that need to be evaluated properly.

Response 5: it is true, the data don’t follow any isochrion trend, suggestive an open system with disturbance by different fluids

6.Please discuss Rb/Sr signatures and Sr isotopic data available in the literature on the crustal rocks from which these pegmatites were sourced and into which they were emplaced, in order to support proposed processes of anataxis/metasomatism by external fluids.

Response 6: actually there is Rb/Sr signatures and Sr isotopic data from the crustal rocks in Nanping pegmatite field, but it is very simple, only show the 87Sr/86Sr ratio is more than 0.71[Yang et al., 1987], so we don’t cited its data.

7.What decay rate of Rb is used to calculate initial Sr ratios? I don't see it cited anywhere. Please use the most recent and accurate value of λ87Rb 1.397*10−11 by Villa et al 2015. Mention in data Table and/or methods.

Response 7: we cited the Ref., thank you.

8.Finally, please have the manuscript proof-read for typos and correct syntax prior to resubmitting. I attached an annotated pdf which I commented on while reading. Please excuse some repetition of comments above. 

Response 8: thank you very much, based on your annotated version, we check all and modified it.

Reviewer 2 Report

Dear Editor,

Dear authors,

This manuscript presents a study of mineralogy and geochemistry of Sr-bearing Nanping Nº 31 Pegmatite (SE China) and the implications for Sr circulation and post-magmatic processes in granitic system. The authors conclude to a magmatic-hydrothermal origin of Sr enrichment. There are more studies on this interesting topic, however, this manuscript may be of interest to the community of economic geologists.

I have some criticisms regarding the way of presenting the results that make it difficult to interpret the textures and chemical compositions of minerals with Sr to discuss their magmatic vs hydrothermal origin. The most important problems you must address are:

Geologic Background and sampling

The first Paleozoic tectono-thermal events resulted in a partial melting of the continental crust and generated type S granites with peak ages from 430 to 400 Ma. Does the pegmatite studied with a columbite dated 387 Ma belong to this type of magmatism? What is the age of orogenic metamorphism in the region?

The mineralogical composition of each zone is very simplified because only the main minerals are indicated (lines 80 to 86). Later in the text there are minerals that are not included in this description. P.e. K Feldspar is present in all five zones (lines 91-93)

Results

Based on your descriptions, it is difficult to understand what is primary and what is secondary i.e. what resulted from magmatic crystallization and whether what you interpret as secondary is due to post-magmatic change or "primary" crystallization from a hydrothermal fluid.

It is very difficult to follow the deposition sequence of the different minerals and if they result from the alteration of previous minerals. What was the magmatic mineral association?

I suggest that when there are several generations of the same mineral it is referred to for example in the following way Albite I and Albite II (where one is magmatic and the other results for example from an albitization process). Mineralogical and textural observations are essential to propose an interpretation of the chemical data and a formation scenario for the studied deposit.

A figure should be presented with the crystallization sequence and indicating the magmatic and post-magmatic stage and the different hydrothermal alteration stages.

In my opinion and based on the presented data, the manuscript requires a revision.

I have attached your manuscript with edits, comments and suggestions to improve the text and the discussion of your data.

Author Response

1.The first Paleozoic tectono-thermal events resulted in a partial melting of the continental crust and generated type S granites with peak ages from 430 to 400 Ma. Does the pegmatite studied with a columbite dated 387 Ma belong to this type of magmatism? What is the age of orogenic metamorphism in the region?

Response 1: we add the age of orogenic metamorphism in the region, and clearly state the 387 Ma belong to this type of magmatism for the Nanping No. 31 pegmaite.

2.The mineralogical composition of each zone is very simplified because only the main minerals are indicated (lines 80 to 86). Later in the text there are minerals that are not included in this description. P.e. K Feldspar is present in all five zones (lines 91-93)

Response 2: there is a mistake, K-Feldspar is present in all five zones (lines 91-93), is present in all type pegmatites in Nanping. Here we modified it.

3.Based on your descriptions, it is difficult to understand what is primary and what is secondary i.e. what resulted from magmatic crystallization and whether what you interpret as secondary is due to post-magmatic change or "primary" crystallization from a hydrothermal fluid.

Response 3: in pegmatite, one mineral may occur as primary and secondary phases distributed in different associations. Primary minerals crystallized during the magmatic stages, secondary phase formed under the hydrothermal fluids

4.It is very difficult to follow the deposition sequence of the different minerals and if they result from the alteration of previous minerals. What was the magmatic mineral association?

Response 4: we add another table to show parts of minerals and their schematic sequence in the Nanping No. 31 pegmatite

5.I suggest that when there are several generations of the same mineral it is referred to for example in the following way Albite I and Albite II (where one is magmatic and the other results for example from an albitization process). Mineralogical and textural observations are essential to propose an interpretation of the chemical data and a formation scenario for the studied deposit.

Response 5: in the Nanping No. 31pegmatite, albite has several generations in zones I, II, III, and IV, which are magmatic phase, but the secondary albite occurs as one of the alternation products of K-feldspar in zones IV and V.

6.A figure should be presented with the crystallization sequence and indicating the magmatic and post-magmatic stage and the different hydrothermal alteration stages.

Response 6: we add another table to show the crystallization sequence of the related minerals

7.In my opinion and based on the presented data, the manuscript requires a revision. I have attached your manuscript with edits, comments and suggestions to improve the text and the discussion of your data.

Response 7: thank you very much for your constructive comments/suggestions.

Reviewer 3 Report

In this paper, the authors focused on the petrography, chemical composition, and Sr isotope of Sr-bearing phosphates from the Nanping No. 31 pegmatite in southeastern China to characterize post-magmatic stages and geochemical recirculation of Sr in granitic systems. The detailed geochemical data reported provide new insights on the chemical composition of the samples, the occurrence and distribution of Sr in granitic rocks.

The methods have been applied correctly and the multi-methodological approach provide a good viewpoint on the geochemical similarities and differences of minerals in the granitic system. The experimental protocols and discussion of data are well detailed. Overall the quality of the study is high. Thus, I suggest the publication of the paper with minor revision. Please consider the comments below to revise your manuscript:

Line 49. In the sentence beginning “In this study,…”,  remove “therefore”

Line 75. SN direction seems odd. Please revise.

Line 82. What does LCT stand for?

Line 102. What does ZAF stand for?

Line 110. A spot size of 90. The unit of measurement is missing.

Line 325. Change “pegmaite” with “pegmatite”

Author Response

Line 49. In the sentence beginning “In this study,…”,  remove “therefore”

Done

Line 75. SN direction seems odd. Please revise.

Done, SW

Line 82. What does LCT stand for?

Done

Line 102. What does ZAF stand for?

ZAF is special program for EPMA, Z = Atomic number (Z) correction; A= Absorption correction; F= Fluorescence correction.

Line 110. A spot size of 90. The unit of measurement is missing.

Done

Line 325. Change “pegmaite” with “pegmatite”

Done

Thank you very much for your constructive comments and suggestions

Round 2

Reviewer 1 Report

The paper has improved considerably by the inclusion of a paragenetic diagram, the description of the different pegmatite zones, and the inclusion of a Sr data plot. I am happy with the replies and appreciate the short turnaround time for this revision. However, I still don't think the analyses of the Sr data is entirely sound, and find that further editing of the text is required.

Two examples of incorrect interpretation/discussion of the Sr isotope data:

  • Line 365: The 87Sr/86 ratios of secondary apatites range from 0.72747 to 0.73357, suggesting that secondary apatites crystallized under different degree of rediogenic fluids through the whole hydrothermal stages of the pegmatite. Therefore, the occurrence, chemical composition and isotopic features of Sr-bearing minerals reflect a significant chemical disturbance of Rb-Sr system by  hydrothermal fluids from the wall-rocks in an open system”

This needs rewriting, fluids are not radiogenic. The fluids, depending on what rocks they interacted with, can have different Sr isotopic signatures. If the wall-rock is older/more Rb rich than the pegmatite, its Sr isotopic signature will be more radiogenic, and fluids derived from them can then impart a more radiogenic signature onto the secnoddary minerals that crystallize from them.

Where does the wall-rock come in here?  The Sr isotopic signature of the wall-rock (being what again?) has not been discussed yet.  If the authors expect the wall-rock to produce fluids with radiogenic Sr signatures (i.e.  above the expected initial of the pegmatite, also not discussed), that should be explained/discussed before you can derive at the conclusion that ' significant disturbance by hydrothermal fluids from the wall-rocks in an open system'.

  • Line 388: The 87Rb/86Sr and 87Sr/86Sr ratio of secondary Sr-bearing phosphates (strontiohurlbutite, bertossaite and palermoite), illustrated a small amount radiogenic 87Sr by the decay of Rb.

This is incorrect. All 87Sr is radiogenic by the decay of Rb. Maybe the authors are trying to distinguish between radiogenic ingrowth from 87Rb within the crystal since crystallisation, or inherited 87Sr (already formed by decay of 87Rb prior to crystallisation)?

I am also a bit unsatisfied with the suggestion there are no solid solution/compositional trends when for example the two ‘representative’ analyses shown for palermoite, clearly show significant variations in Sr and Ca. If hundreds of points have been analysed, it would be good practice to show the compositional variability within each of these mineral groups in the form of data plots (plotting concentrations of elements assumed to compete for the same site f.e.) and by including in the table along with the ‘representative compositions’, average compositions plus the standard deviation, for n number of analyses.

Cation substitution schemes are proposed in Section 5.1  (Lines 304-315), but this could/should be fleshed out by showing us plots to demonstrate compositional trends of Sr vs Ba vs Ca contents etc in the different mineral groups (since so many analyses were made).

Discussion:

The discussion would make more sense if first the petrogenesis is discussed (i.e. What is the inferred magmatic initial Sr signature of the pegmatite?), and then discuss how the Sr isotopes are affected by hydrothermal fluids, and specifically which mineral groups have been affected to which degrees.

Line 378:  Why does an average 0.729 Sr ratio suggest an abyssal type pegmatite in an open system? Can the authors explain and cite Sr ratios typical of abyssal type pegmatites? (References?)

Line 367: The Nanping No. 31 pegmatite is formed by anatexis in lower crust [20]

The authors here cite a previous study of theirs, indicating that the crustal anataxis was derived from findings of previous studies, yet the abstract and conclusions suggest that this is a finding based on the Sr isotopes signatures presented here.

I previously commented the following: The authors propose that the pegmatite formed by crustal anataxis followed by alteration by wall-rock derived external fluids, without considering the isotopic signatures of those reservoirs. In other words, the conclusions are not supported by the data.

Although slightly reworded in the discussion, this comment has not been addressed. The Sr data are so heavily disturbed by late-magmatic/hydrothermal overprint (open system behaviour) that their primary magmatic Sr signature is completely lost. Hence, in no way does the Sr data allude to petrogenetic history that would hint at lower crustal anatexis. Please redact abstract an conclusions accordingly.

If the authors want the Sr isotope study to support this earlier finding (of an origin by crustal anataxis), then the authors need to discuss what they expect the primary Sr isotopic signature to be for the lower crust and the melts derived from them (the pegmatite). Or please clarify throughout that this inference is based on previous studies, and on what type of geochemical data that is based.  At present, the Sr isotopes do not say anything about the petrogenesis of the pegmatite.

I have attached the pdf, with further annotations and highlights (major comments annotated in the pdf are copied above).

Best regards

Author Response

Thank you very much for your constructive suggestions!

  1. Line 365: “The 87Sr/86 ratios of secondary apatites range from 0.72747 to 0.73357, suggesting that secondary apatites crystallized under different degree of rediogenic fluids through the whole hydrothermal stages of the pegmatite. Therefore, the occurrence, chemical composition and isotopic features of Sr-bearing minerals reflect a significant chemical disturbance of Rb-Sr system by  hydrothermal fluids from the wall-rocks in an open system”

Response 1: according to your suggestions, we have rewritten, please check the new version.

  1. Line 388: The 87Rb/86Sr and 87Sr/86Sr ratio of secondary Sr-bearing phosphates (strontiohurlbutite, bertossaite and palermoite), illustrated a small amount radiogenic 87Sr by the decay of Rb.

Response 2: we have checked it, the statement is wrong.

  1. I am also a bit unsatisfied with the suggestion there are no solid solution/compositional trends when for example the two ‘representative’ analyses shown for palermoite, clearly show significant variations in Sr and Ca. If hundreds of points have been analysed, it would be good practice to show the compositional variability within each of these mineral groups in the form of data plots (plotting concentrations of elements assumed to compete for the same site f.e.) and by including in the table along with the ‘representative compositions’, average compositions plus the standard deviation, for n number of analyses.

Response 3: Yes, there some solid solutions, such as hurlbutite vs strontiohurlbutite, bertossaite vs palermoite etc. Since most of these minerals occur as secondary phases in the Nanping pegmatite, it is not necessary to describe it here, but some of solid solutions can be found in our previous paper [Rao et al. 2017].

  1. Cation substitution schemes are proposed in Section 5.1  (Lines 304-315), but this could/should be fleshed out by showing us plots to demonstrate compositional trends of Sr vs Ba vs Ca contents etc in the different mineral groups (since so many analyses were made).

Response 4: good suggestion, but there are four groups minerals about the compositional trends of Sr vs Ba vs Ca contents. If we plot it here, there more Figures.

  1. The discussion would make more sense if first the petrogenesis is discussed (i.e. What is the inferred magmatic initial Sr signature of the pegmatite?), and then discuss how the Sr isotopes are affected by hydrothermal fluids, and specifically which mineral groups have been affected to which degrees.

Response 5: good suggestion, we try to do it.

  1. Line 378:  Why does an average 0.729 Sr ratio suggest an abyssal type pegmatite in an open system? Can the authors explain and cite Sr ratios typical of abyssal type pegmatites? (References?)

Response 6: have done

  1. Line 367: The Nanping No. 31 pegmatite is formed by anatexis in lower crust [20]

Response 7: have done.

  1. Although slightly reworded in the discussion, this comment has not been addressed. The Sr data are so heavily disturbed by late-magmatic/hydrothermal overprint (open system behaviour) that their primary magmatic Sr signature is completely lost. Hence, in no way does the Sr data allude to petrogenetic history that would hint at lower crustal anatexis. Please redact abstract an conclusions accordingly.

Response 8: have done.

  1. If the authors want the Sr isotope study to support this earlier finding (of an origin by crustal anataxis), then the authors need to discuss what they expect the primary Sr isotopic signature to be for the lower crust and the melts derived from them (the pegmatite). Or please clarify throughout that this inference is based on previous studies, and on what type of geochemical data that is based.  At present, the Sr isotopes do not say anything about the petrogenesis of the pegmatite.

Response 9: have modified accordingly.

  1. I have attached the pdf, with further annotations and highlights (major comments annotated in the pdf are copied above).

Response 10: thank you very much again.

Reviewer 2 Report

Dear Editor,

Dear authors,

As already mentioned in this manuscript it presents a study of mineralogy and geochemistry of Nanping No. 31 Pegmatite (SE China) and the implications for the circulation of Sr and post-magmatic processes in the granitic systems and is of interest to the community of economic geologists.

The suggestions I made for the presentation of the results were accepted and in my opinion it became clearer and easier to interpret textures and chemical compositions of minerals with Sr to discuss their magmatic vs hydrothermal origin.

In my opinion, the revised manuscript can be accepted for publication.

Author Response

Thank you very much for your constructive suggestions!